# FLOMPY: An Open-Source Toolbox for Floodwater Mapping Using Sentinel-1 Intensity Time Series

**Kleanthis Karamvasis ***  **and Vassilia Karathanassi**

Laboratory of Remote Sensing, National Technical University of Athens, 9 Iroon Polytechniou Str., Zographos, 15780 Athens, Greece; karathan@survey.ntua.gr
* Correspondence: karamvasisk@central.ntua.gr

**Abstract:** A new automatic, free and open-source python toolbox for the mapping of floodwater is presented. The output of the toolbox is a binary mask of floodwater at a user-specified time point within geographical boundaries. It exploits the high spatial (10 m) and temporal (6 days per orbit over Europe) resolution of Sentinel-1 GRD intensity time series and is based on four processing steps. In the first step, a selection of Sentinel-1 images related to pre-flood (baseline) state and flood state is performed. In the second step, the preprocessing of the selected images is performed in order to create a co-registered stack with all the pre-flood and flood images. In the third step, a statistical temporal analysis is performed and a *t*-score map that represents the changes due to a flood event is calculated. Finally, in the fourth step, a classification procedure based on the *t*-score map is performed to extract the final flood map. A thorough analysis based on several flood events is presented to demonstrate the main benefits, limitations and the potential of the proposed methodology. The validation was performed using Copernicus Emergency Management Service (EMS) products. In all case studies, overall accuracies were higher than 0.95 with Kappa scores higher than 0.76. We believe that the end-user community can benefit by exploiting the flood maps of the proposed methodological pipeline by using the provided open-source toolbox.

**Keywords:** flooding; time series; Sentinel-1; thresholding; open-source software

## 1. Introduction

Mapping the spatial extent of surface waters is considered an important step for many initiatives related to water sustainability and natural hazards such as floods [1]. Floods are among the most severe natural disasters, causing great losses to anthropogenic and natural environment around the globe [2]. Moreover, it is important to improve flood relief efforts due to expected increases in the frequency and magnitude of flood events due to climate change [3]. Fast responses from decision makers and emergency managers can mitigate casualties and damages. For this reason, near-real time spaceborne remote sensing data could be exploited in order to provide accurate and rapid maps of affected area by floods [4]. These maps can be used for calibration and validation of hydrological models [5]. Furthermore, they can help to better set intervention priorities in order to form a loss mitigation plan [6] and even improve flood forecasting [7].

Optical multispectral satellite data can be used in an easy and straightforward way in order to identify flood features in a scene [8,9]. However, optical data are not optimal to use mainly due their sensitivity to high cloud coverage which is usually present during rain/flood events [10]. On the other hand, synthetic aperture radar (SAR) sensors have their own source of illumination and can acquire data day and night in all-weather conditions. Over open water surfaces and open floodwater regions, the active SAR pulses have low signal returns due to the specular backscattering [11]. However, inundated vegetation and flooding in urban regions may have strong signal returns due to "double bounce" effect [10]. Moreover, wind can increase the roughness of water surfaces which results to higher backscattering values [12].

A considerable number of methodological approaches based on SAR data have been developed in the recent years. Some of the developed methods are based on supervised approaches that require human intervention [13] and/or they are using labelled data which require manual work. This is an important obstacle for developing a robust automatic approach and an ongoing effort to exploit data-driven approaches is present [14]. On the other hand, unsupervised methods are suitable from automatic workflows. The majority of the unsupervised methods include thresholding operations. Thresholding approaches rely on selecting an adequate threshold in order to determine flooded versus non-flooded areas. Thresholding can be applied on a single backscatter image [15] or on a combination of backscatter time series [16] and on a global or local scale in order to account backscatter spatial variability [17].

It is known that thresholding approaches have many drawbacks and can be affected by a lot of errors [4,12,17] such as (a) Bragg scattering due to wind [18]; (b) "double bounce" effects due to urban and inundated vegetation; (c) water-like features (smooth surfaces, soils with high soil moisture, shadow regions) [19]; (d) speckle effect and (e) selection of threshold. Due to the continuous increase of the data volume and availability, one of the most promising ways to overcome some of the abovementioned drawbacks is via exploiting multi-temporal information [4,20,21]. Change detection approaches exploit the multi-temporal information by comparing data related with pre-flood and flood state. It is important to state that, the forming of the pre-flood (baseline) dataset is a crucial step to accurately estimate the extent of the floodwater using change detection approaches [11,22].

In this work we propose a robust methodological approach able to mitigate the abovementioned drawbacks and provide rapid and accurate flood maps. The proposed approach includes four steps: (a) baseline (pre-flood) dataset formation; (b) preprocessing; (c) SAR statistical temporal analysis; and (d) floodwater classification. The proposed pipeline is provided as a free open-source python tool [23] inspired by the free policy of Sentinel-1 data. The objectives of this work are to release a tool able to produce floodwater maps which mainly can be used for (a) flood damage assessment for fast responses, (b) calibration and validation of hydrological models, and (c) flood forecasting.

This paper is organized as follows. In Section 2, the proposed methodology is introduced. In Section 3, the first experimental results and their validation are presented. An accuracy assessment of the proposed methodology using EMS products as reference data and a comprehensive discussion regarding the theoretical concepts of the proposed methodology and its limitations are presented in Section 4. In Section 5, the conclusions of this study are outlined are presented.

## 2. Methodology

In this section, we describe the proposed methodology that is implemented in Flood Mapping Python toolbox (FLOMPY) [23]. FLOMPY is a free and open-source toolbox which is written in Python 3 and published under the GNU GPL version 3 license. Some modules are based on Sentinel's Application Platform (SNAP) [24] functionalities. The proposed and implemented methodology consists of following steps:

### 2.1. Baseline Dataset Formation

In the first step, a selection of the Sentinel-1 images in order to form a pre-flood (baseline) dataset is performed. Starting with a predefined area of interest and flood event time, all available metadata for Sentinel-1 acquisitions are assessed. We select the optimal orbit track for analysis based on the time difference between acquisition time and the selected time of the flood event. For facilitating the description of the methodology, the acquisition right after the selected time of the flood event will be referred to as the flood image. Next, we form a pre-flood (baseline) dataset by using only the acquisitions with the same orbit as the flood image of the latest past months that are related with low precipitation.

In particular, based on historical precipitation data (ERA-5), we keep only the acquisitions that have previous 5 day cumulative precipitation lower than a predefined threshold (45 mm in most cases). A 3-month time period was selected to minimize the temporal changes which are not related with the flood and to achieve an adequate number of acquisitions (>5) for temporal statistical analysis. Next, the preprocessing step of the selected pre-flood images and the flood image is performed.

*2.2. Preprocessing*

The pre-processing workflow is based on ESA Sentinel's Application Platform (SNAP) functionalities and consists of the following steps [24,25]:

- Orbit correction using precise orbit information or restituted orbit information if the former is not available.
- Thermal noise removal operation.
- Border noise removal which masks artificially low backscatter pixel and invalid data found at the edge of the image swath.
- Radiometric calibration to produce unitless backscatter intensity.
- Subsetting at given spatial extent.
- Co-registration between all images in order to create a stack that will be used for time series analysis.
- Terrain geocoding using the 1-arcsec digital elevation model (DEM) from the Shuttle Radar Topography Mission (SRTM).
- Local incidence angle normalization based on [26].

*2.3. SAR Statistical Temporal Analysis*

After the preprocessing, a statistical temporal analysis between pre-flood (baseline) stack and flood image is performed. In order to exploit the polarimetric capabilities of Sentinel-1 products, both VV and VH polarizations were exploited. In particular, the product of VH and VV polarizations was calculated using formula (1) and used for the temporal statistics (Appendix A). This way, the processing cost was reduced because only one band (product of VV with VH) is used.

$$\sigma_{VV*VH} = 10 log_{10}(\sigma_{VV} \cdot \sigma_{VH}), \tag{1}$$

where, $\sigma_{VV*VH}$, the product of backscatter coefficients of VV and VH in decibel and $\sigma_{VV}$, $\sigma_{VH}$ the unitless backscatter coefficient of VV and VH, respectively.

The temporal *t*-score was selected due to the small sample size and is calculated for each pixel according to the formula (2). The hypothesis is that the backscatter coefficient of the pre-flood state follows a T-distribution. T-distribution is a type of normal distribution for smaller sample sizes.

$$t_{score} = \frac{\sigma_{VV*VH,flood} - mean\sigma_{VV*VH,baseline}}{\frac{std\sigma_{VV*VH,baseline}}{\sqrt{n}}}, \tag{2}$$

where, $\sigma_{VV*VH,flood}$ the product of backscatter coefficients between VV and VH of the flood image.

$mean\sigma_{VV*VH,baseline}$ and $std\sigma_{VV*VH,baseline}$ the average value and the standard deviation of the product of VV and VH backscatter coefficients of the baseline (pre-flood) stack, respectively.

$n$ the number of the acquisitions in the baseline stack.

*2.4. Classification*

In the final step, a novel spatially adaptive thresholding methodology was developed for the classification of the calculated *t*-score map in flooded and non-flooded areas. The

*t*-score has the same resolution with Sentinel-1 GRD products. The classification scheme consists of five main steps (Figure 1D):

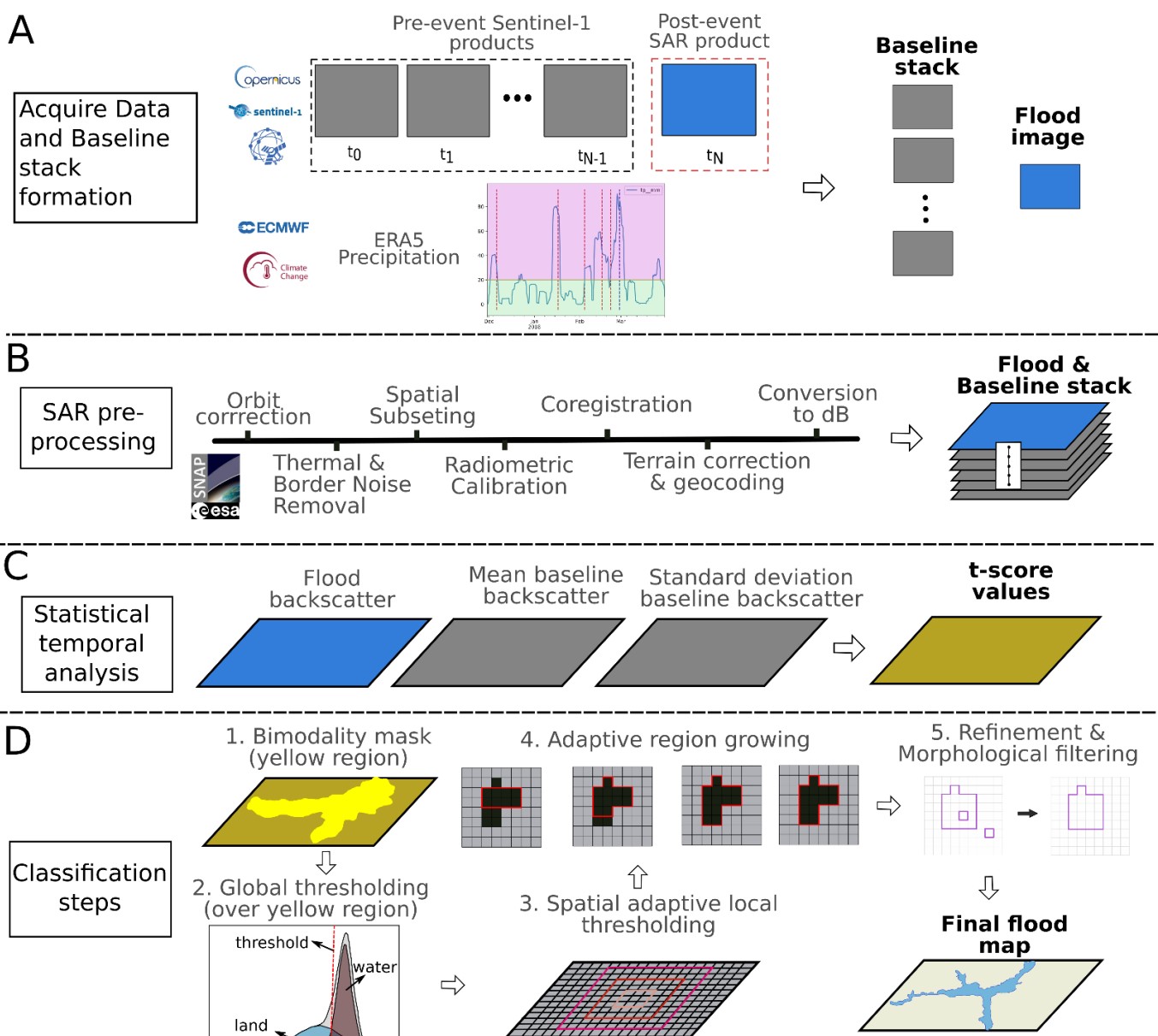

**Figure 1.** Flowchart of the four-step proposed methodological pipeline (FLOMPY).

1. Calculation of a mask in order to achieve similar class sizes of the flooded and non-flooded samples. The use of this mask by histogram thresholding algorithms improves their performance significantly [27,28]. This mask is referred as bimodality mask because it is calculated by using bimodality coefficient (BC) [29]. The bimodality coefficient is expressed by the formula:

$$BC = \frac{s^2 + 1}{k + 3\frac{(n-1)^2}{(n-2)(n-3)}}, \qquad (3)$$

where, s the skewness of the distribution, *k* its kurtosis and *n* the number of the samples.

The bimodality mask is computed as follows. First, BC values are calculated at multiple grid sizes that range from 25 to 500 pixels with a step of 25 pixels. The result of each grid size is saved at the *t*-score's pixel level. Then, a mean BC value for each pixel is calculated and a single thresholding approach is applied. If the BC value is smaller than 0.555, a uniform distribution is expected. Higher values point toward multimodality [29].

2.  In this step, a single threshold that discriminates the flooded and the non-flooded regions based on Kittler-Illingworth algorithm [30] was implemented over the masked *t*-score. The result of this step is a first binary flood mask by applied the scene-level threshold.

3.  Next, a spatial adaptive local thresholding was applied to compress over-detection and under-detection issues from the single scene-level thresholding. The main steps that apply for each flooded-pixel according to the binary flood mask of the previous step are the following:

    a.  For each flooded pixel we select the optimal window size based on the bimodality index value. In particular, from a predefined range of window sizes (from 15 to 80 pixels with a step of 5 pixels) we select the one that yields the maximum bimodality index. If its value is above 0.555 [29] we proceed to the next step. Otherwise, we move in the next flooded pixel.

    b.  A local histogram thresholding based on Otsu algorithm is performed. More information regarding Otsu algorithm can be found in [31,32]. Otsu algorithm was selected due to its good performance in tiled thresholding [28]. From the local thresholding we get two samples. The first sample (sample 1) is the one that includes values less than the local threshold and the second sample (sample 2) in the one that includes values bigger or equal with the local threshold. Next, we test how sample 1 and sample 2 are related with floodwater and non-floodwater population from the step-2 binary flood mask by calculating the following flags:

        i.   Water_flag1: Is sample1 similar to floodwater population.
        ii.  Water_flag2: Does sample 1 has significantly lower values in respect to floodwater population.
        iii. Land_flag1: Is sample 2 similar to non-floodwater population.
        iv.  Land_flag2: Does sample 2 has significantly higher values in respect to floodwater population.

    The similarity flags (Water_flag1, Land_flag1) were calculated using a non-parametric two-sample Kolmogorov-Smirnov test with 95% confidence ratio [33]. The comparison flags ((Water_flag2, Land_flag2) were calculated by utilizing the Fisher discriminant ratio [34]. In order to ensure that the local thresholding can improve the scene-level thresholding we follow the decision tree (Figure 2) based on the computed flag values.

4.  Region growing was used in order to improve pixel-based performance by accounting spatial information, compress under-detection issues and reduce omission errors [11]. The region growing algorithm that we developed searches for pixels within the *t*-score map that are connected neighbours to the flood pixels (seeds) from step 3 and that fall within a tolerance criterion. We point out that the tolerance level is not the same for each seed, but it is linearly related with its *t*-score value. This way the algorithm is stricter for seeds with higher *t*-scores than for seeds with lower *t*-scores.

5.  Refinement is the final step and consists of (a) masking out high slope regions (>15 degrees) from available DEM and (b) morphological filtering based on a predefined minimum mapping unit. In particular, based on the predefined minimum mapping unit we fill the small holes, and we removed the small objects.

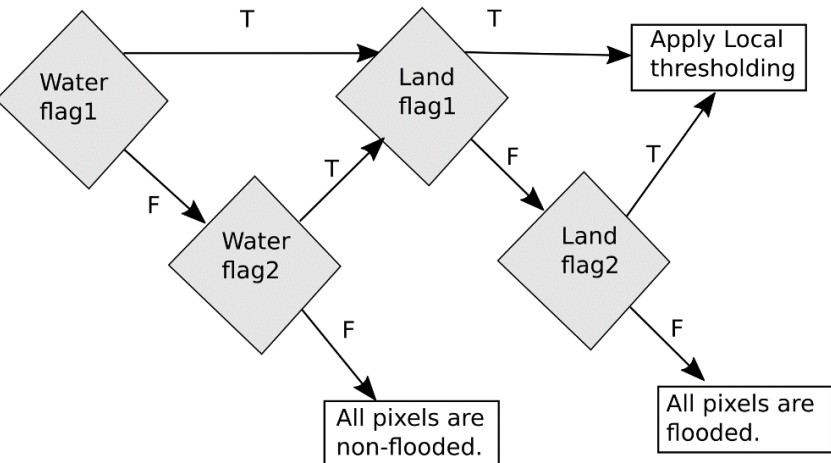

**Figure 2.** Decision tree of adaptive local thresholding methodology.

**3. First Experimental Results: The Case Study of Pinios Flood**

*3.1. Area of Interest*

The algorithm was initially implemented for mapping the Pinios flood. Pinios catastrophic flood event occurred in spring of 2018 and was caused by a series of storm events from 21 to 26 February 2018 [35,36]. In the particular flood event, Pinios River and all its tributaries have overflowed since 24 February 2018 and hundreds of acres of rural and urban areas have been affected by flooding [37].

The area of interest (AOI) is located in the region of central Greece between the cities of Larissa and Trikala. (Figure 3a). The complex dendritic hydrographic network consists of Pinios River and its tributaries, and has a length of around 1188 km [38]. Due to the high frequency of the flood events along Pinios River, many flood protection works have been performed in the past and still planned for the future [39].

We focus on a region (Figure 3b) that covers an area of about 764 $km^2$ and with elevations that vary from 69 to 970 m above sea level (a.s.l.). The relative low terrain elevation along the hydrographic network increases the flood risk at the study area [38]. In Figure 3c, a map of maximum floodwater depth is shown to demonstrate the severity and the impact of the flood events over the area of interest. According to the land cover map (Figure 3d), cropland covers the largest part of the AOI, especially along Pinios River which is mainly used for irrigation [38,39]. The intense and high-frequency flood events can cause significant damages to the agricultural production and to several villages along the river network.

*3.2. Datasets*

SAR data from short revisiting (6-day over Europe) Sentinel-1 constellation from ESA (European Space Agency) are used. Sentinel-1 consists of two satellites namely, Sentinel-1A and Sentinel-1B that were launched in 2014 and 2016, respectively. Sentinel-1 constellation is a polar-orbiting radar imaging system working at C-band (~5.7 cm wavelength). All the Sentinel-1 TOPS Interferometric Wide swath (IW) acquisitions (9) from the descending track with relative orbit number 80, dating from December 2017 till the flood event (28 February 2018) have been processed. All the data are in the Ground Range Detected (GRD) format at a 10-m spatial resolution and both VV and VH polarizations are considered.

The ancillary data consists of the 1arcsec SRTM Digital Elevation Model (DEM) and the precise orbit ephemerides. For validation purposes the EMS product was used. The EMS product was extracted using the semi-automatic method. More information regarding the semi-automatic method and the quality control of the EMS products can be found in [40]. The EMS flood map was produced from a semi-automatic approach on a 3 m Cosmo-Skymed image acquired at 04:00 of 28 February 2018 [37]. We point out that in this

case study, the Sentinel-1 image that was used as a flood image was acquired at 04:20 of the 28 February 2018.

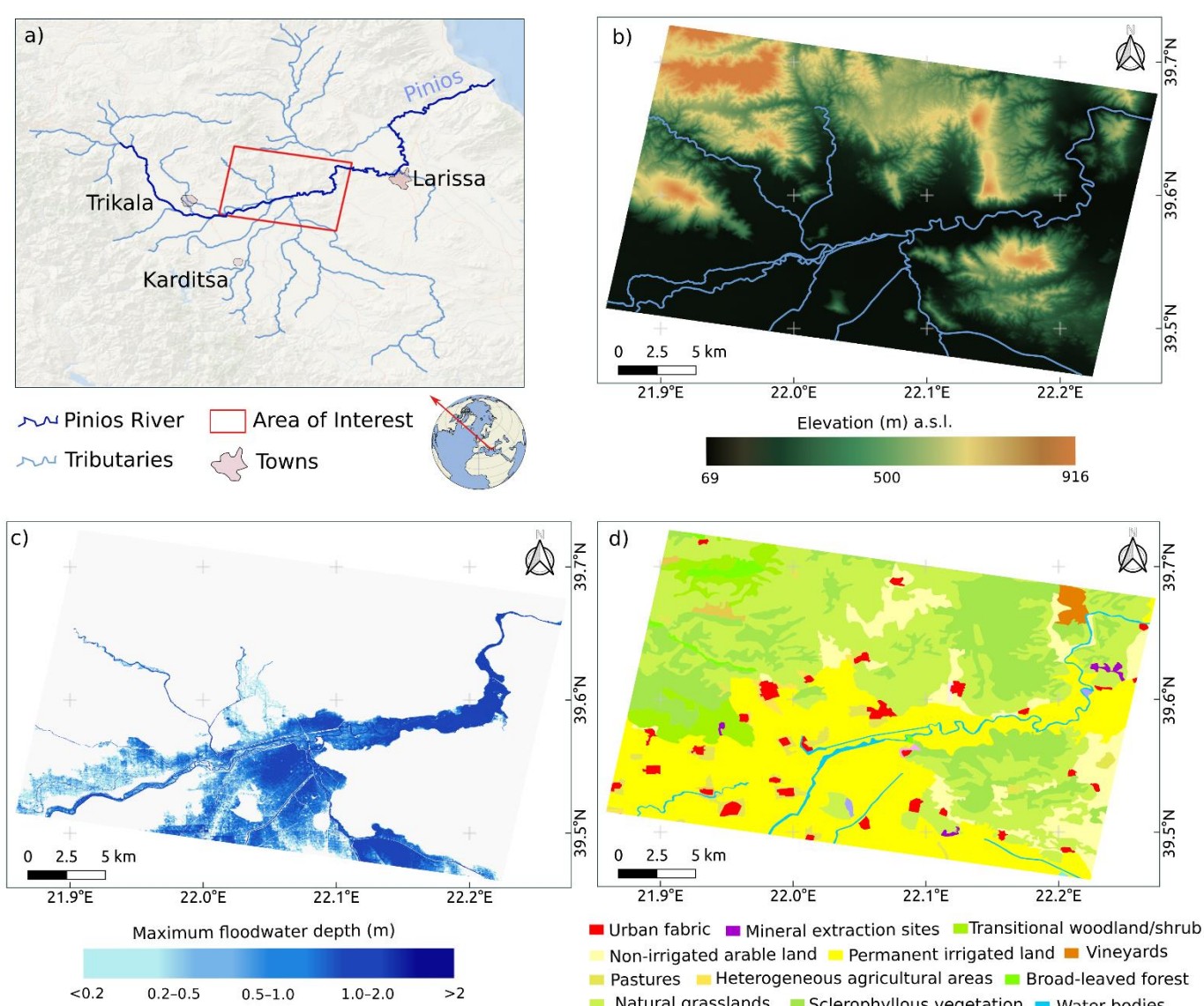

**Figure 3.** Description of the case study (**a**) Pinios river with its major tributaries and main cities over a broader area. The area of interest is denoted with the red rectangle. (**b**) Elevation and hydrographic network of the area of interest (**c**) Spatial distribution of maximum floodwater depth for return period T = 50-years over the area of interest [41]. (**d**) Corine land cover level-2 classes of the study area [42]. Reference system is WGS84. Maps were created using QGIS. QGIS Development Team, 2021. QGIS Geographic Information System (Version 3.16). Open-Source Geospatial Foundation Project. http://qgis.osgeo.org, (accessed on 19 October 2021).

### 3.3. Results and Quality Assessment

In this section the FLOMPY results and the validation with EMS product are presented.

In Figure 4a, an optical image that covers a part of the area of interest is presented. This part will be used for validation purposes. We will focus in three regions to comment on FLOMPY's performance. In Figure 4b, we can visually identify the open floodwater regions with high negative values (dark tones) in the *t*-score map. The high negative values correspond to a big negative change to the backscatter characteristics of these regions that are connected with the change due to flood event. In Figure 4c, we can visually verify the

strong agreement of the FLOMPY results with the dark regions of the *t*-score map and with the EMS product.

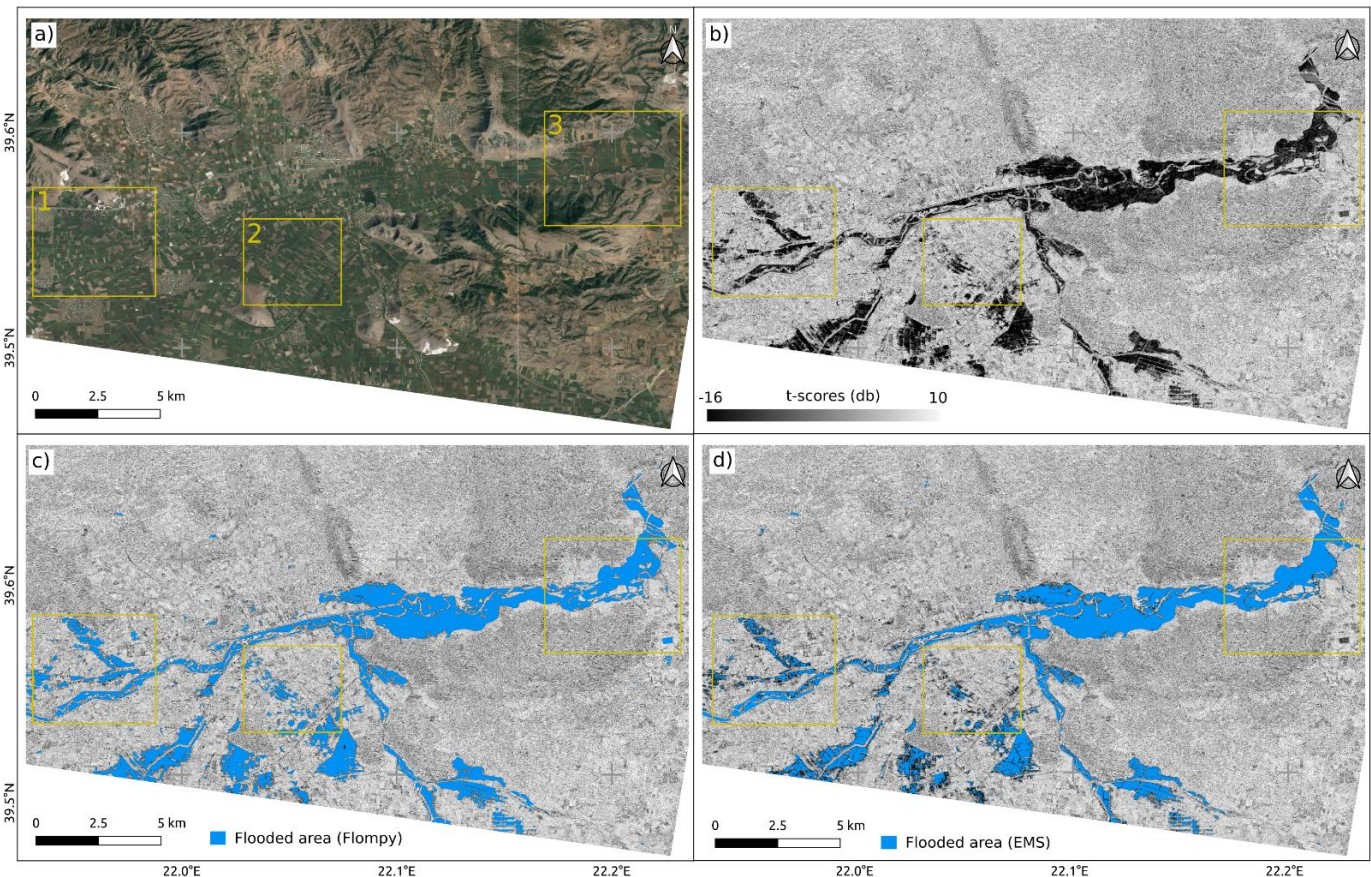

**Figure 4.** Results of the case study (**a**) Optical Google earth imagery. (**b**) T-scores of the Sentinel−1 intensity dataset. (**c**) FLOMPY results overlaid over *t*-score image. (**d**) EMS results overlaid over *t*-score image.

The quantitative accuracy assessment of the FLOMPY results was also performed using EMS product as ground truth. Firstly, EMS vector product was rasterized and resampled to FLOMPY 10-m map using nearest neighbor resampling. Then, the accuracy metrics that were calculated according to [43] are the following:

- Overall accuracy which is the proportion of the flood pixels that are mapped correctly.
- User's accuracy which is the accuracy related to the commission errors of the flood pixels.
- Producer's accuracy which is the accuracy related to the omission errors of the flood pixels
- Kappa score which is a performance metric of the classification compared to just randomly assigning values. Kappa score can range from −1 to 1. Values close to one indicate that the classification has significantly better performance that random.

In Pinios flood case study the overall accuracy was 0.97 with a kappa score of 0.76. The user's accuracy of floodwater was 0.78 and the producer's accuracy of floodwater was 0.76.

We strongly believe that the actual performance of the FLOMPY toolbox cannot be sufficiently depicted by only comparing with EMS products. The abovementioned statement is valid over the three selected regions (Figure 4a) where discrepancies between t-score map and EMS product are visible (Figure 5). Based on *t*-score images, EMS performance varies at each region. We point out that the omission errors of the EMS product affect the derived accuracy metrics and underestimate the actual FLOMPY's performance. However, due to the lack of other validation data, EMS products were considered as ground truth data.

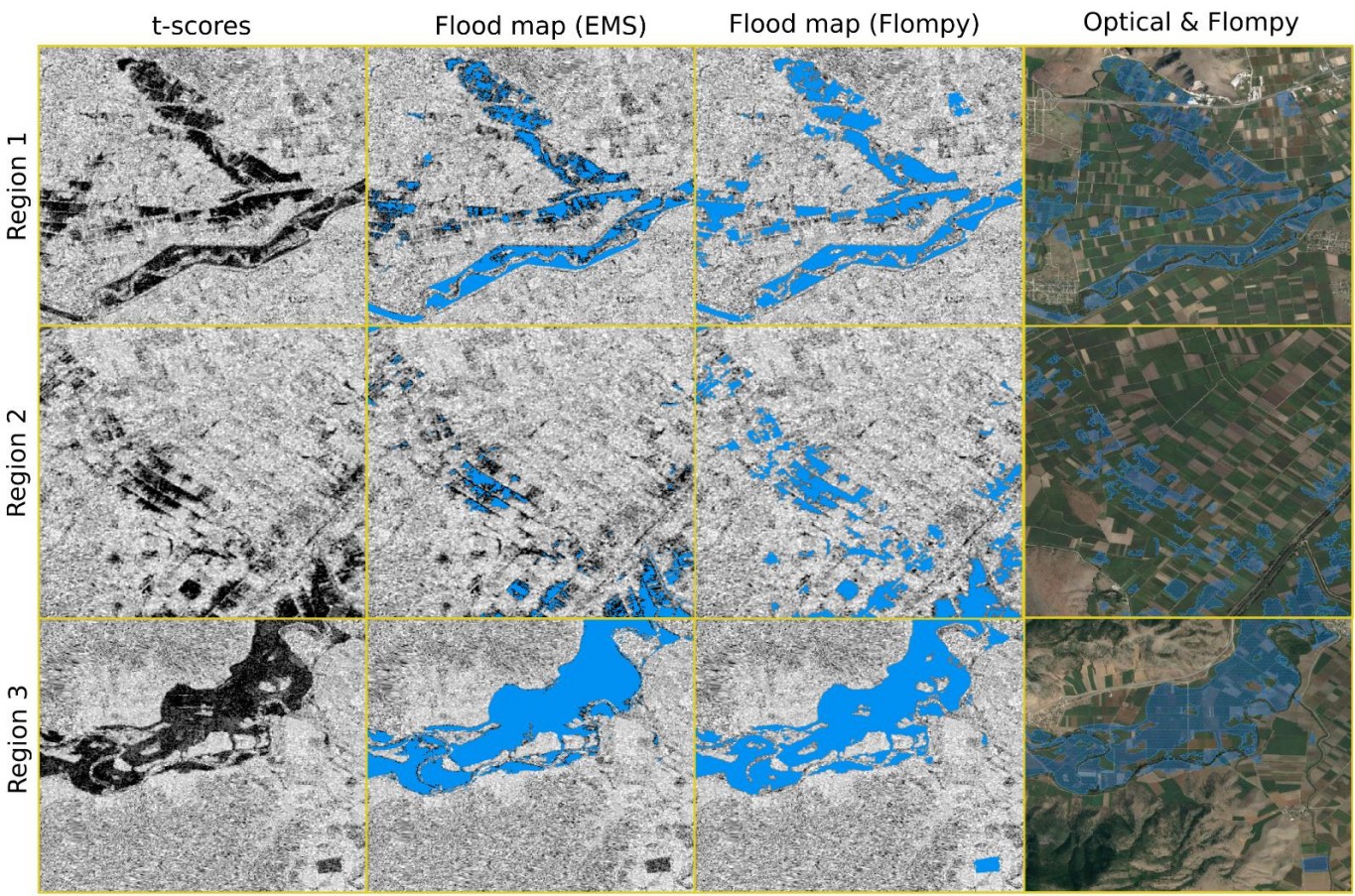

**Figure 5.** Results of selected regions for the case study.

## 4. Validation over Other Case Studies and Discussion

In this section, we describe the main benefits and limitations of the FLOMPY toolbox. FLOMPY toolbox aims at exploiting big EO data and is especially designed for Sentinel-1 intensity data. Based on a selection of the pre-flood images, a statistical temporal analysis and an adaptive thresholding method from the satisfactory results were obtained over Pinios flood case study. Moreover, based on thorough experimentation a fixed parametrization for the FLOMPY pipeline was decided.

In order to test the transferability of the implemented methodology, we assessed the accuracy based on EMS products in four more EMS cases. In order to be consistent, we selected EMS cases that their product is based on the analysis of Sentinel-1 images. The same images were used by FLOMPY toolbox. In Figure 6, the main land cover characteristics for each flood case are presented. We point out that in those cases, the floodwater was found over bare soil and low-height vegetated regions. In the Table 1, the FLOMPY's accuracy metrics are presented. Validation figures for each EMS case can be found in the Supplementary Material. FLOMPY yielded high and consistent accuracies over the presented case studies, which is promising. The SAR temporal *t*-scores proved to be an objective measure of changes due to flood events. Moreover, the pixel-wise implementation of the FLOMPY is compatible for computation cost improvements such as parallelization.

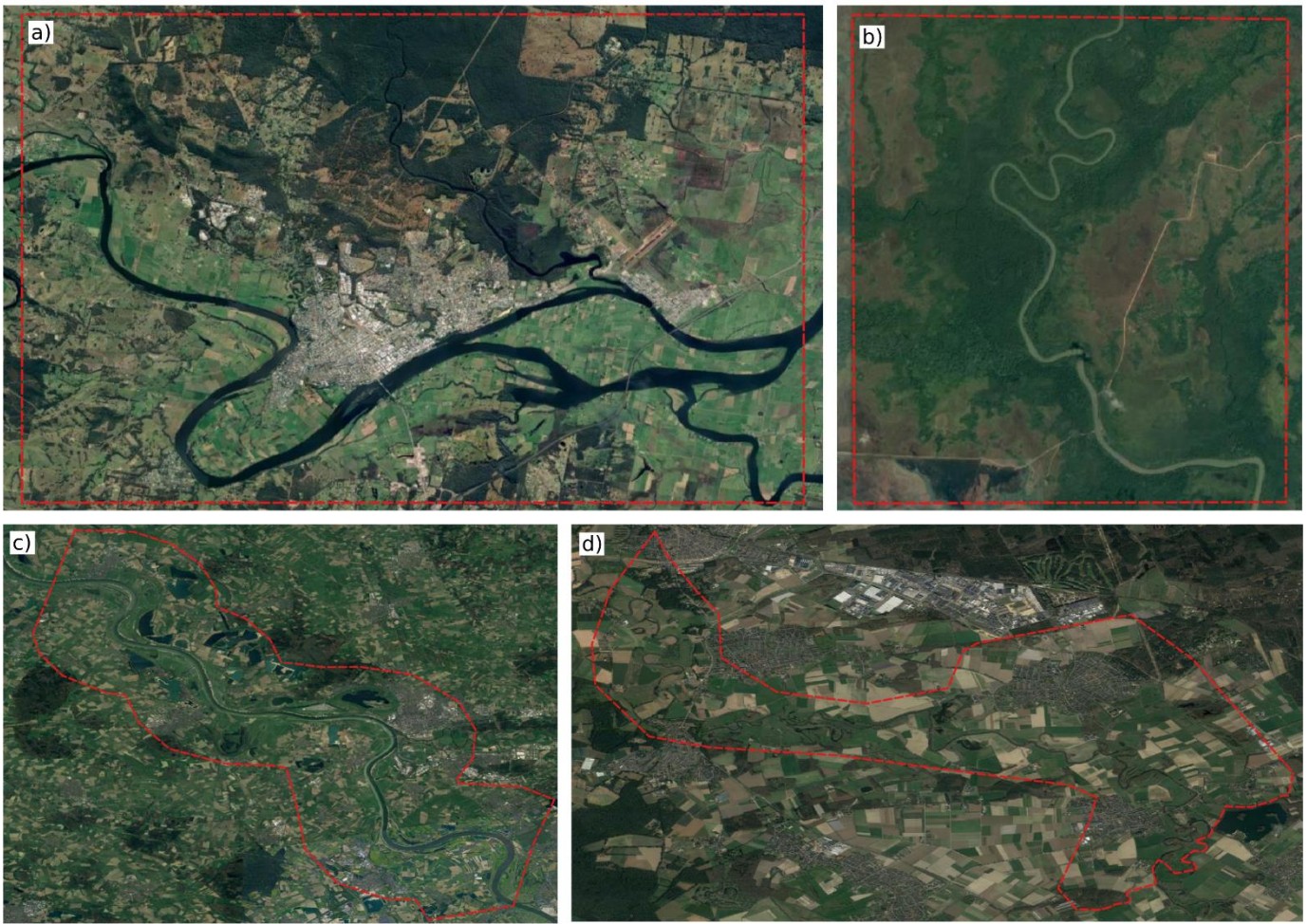

**Figure 6.** Optical Google earth imagery of validation regions over four EMS cases (**a**) EMSR504. (**b**) EMSR456. (**c**) EMSR497. (**d**) EMSR520. The validation regions are denoted with the red polygons.

**Table 1.** Accuracy assessment of FLOMPY products in four EMS cases.

| Case study | EMSR504 | EMSR456 | EMSR497 | EMSR520 |
|---|---|---|---|---|
| Sentinel-1 Datetime | 19 March 2021 19:07 | 16 August 2020 23:50 | 3 February 2021 05:49 | 16 July 2021 05:41 |
| Location | Taree (Australia) | Nicaragua | Germany | The Netherlands |
| Overall accuracy (OA) | 0.95 | 0.97 | 0.98 | 0.95 |
| User's accuracy (UA) | 0.83 | 0.88 | 0.81 | 0.85 |
| Producer's accuracy (PA) | 0.82 | 0.80 | 0.81 | 0.75 |
| Kappa score | 0.80 | 0.83 | 0.79 | 0.77 |

Other important issue is related with the flood classification accuracy. As we already presented in the previous section, EMS accuracy is not homogeneous over different regions of an EMS product. Based on our further analysis of four more EMS cases, we realized that there is also an issue regarding offset errors in EMRS504 [44]. As we can see in Figure 7, a systematic offset of the EMS flood map [44] can be observed. In the same figure, a superior performance of the FLOMPY map can be also observed. To our knowledge, no in-situ observations were available for any of the investigated case studies. Event though, the purpose of this study is not to evaluate existing EMS products, we believe that it is important to focus on producing high-quality validated datasets in order to better assess new developed methodologies.

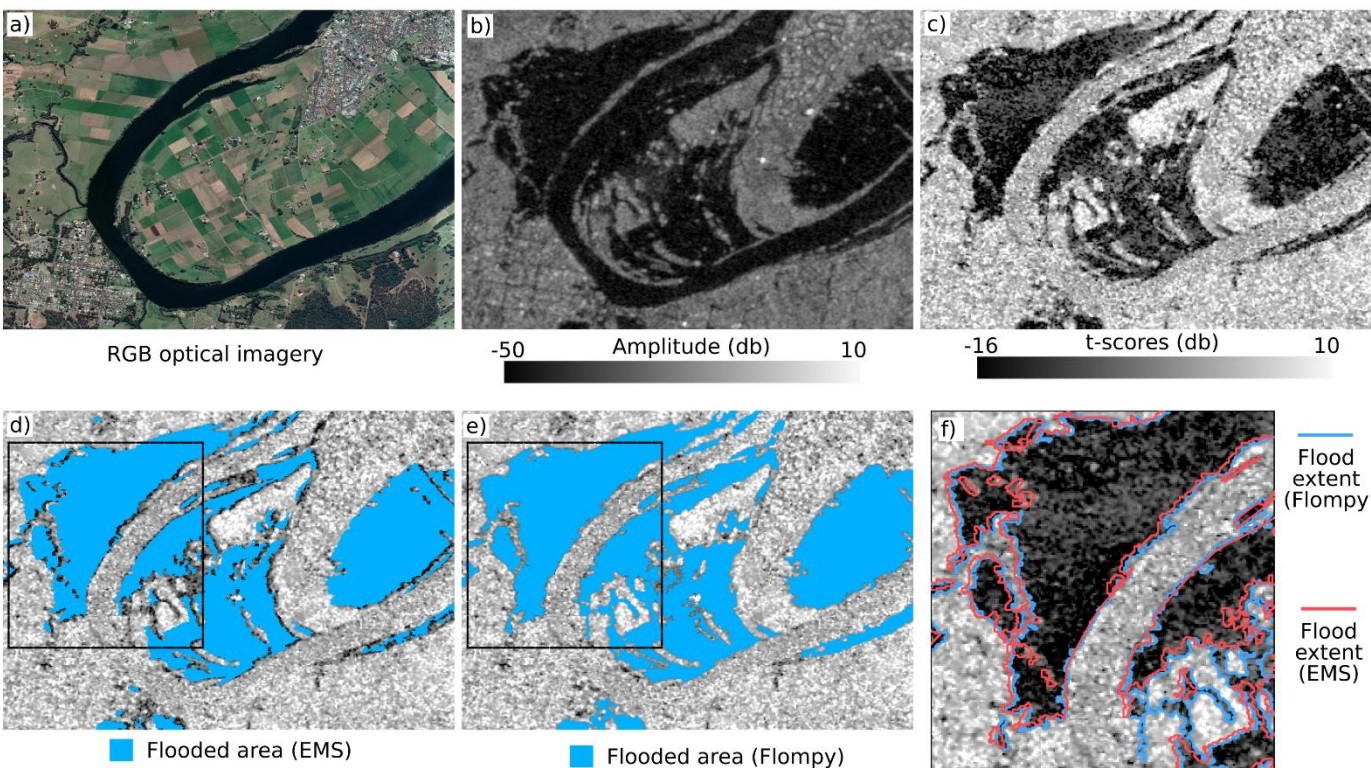

**Figure 7.** Geometric distortions of EMS product (EMSR504) (**a**) Optical Google earth imagery. (**b**) Amplitude which is the product of backscatter coefficients of VV and VH in decibel at flood date. (**c**) T-scores of the Sentinel−1 intensity dataset. (**d**) EMS results overlaid over *t*-score image. (**e**) FLOMPY results overlaid over *t*-score image. (**f**) Flood extents between FLOMPY and EMS products.

One of the biggest limitations of the FLOMPY pipeline is related with the mapping of flooded regions over medium- and high-height vegetated and urban environments. FLOMPY pipeline was specifically designed to map open floodwater over bare soil and low-height vegetated regions where a decrease of backscatter coefficient is distinct due to specular scattering mechanism [11]. Over flooded medium- and high-height vegetated regions, as well as, flooded urban regions a high increase of the backscatter coefficient can also be observed due to double-bounce scattering mechanisms [17]. Over these regions, mapping floodwater in an unsupervised way at Sentinel-1 GRD IW resolution (~20 × 22 m) is considered a really challenging task [45]. One of the future improvements of the FLOMPY toolbox will be the exploitation of interferometric coherence information in order to overcome the abovementioned limitation.

One other limitation of the FLOMPY methodology is related with the quality of the baseline stack. Currently, the baseline stack is formed using images from three months before the flood event. In some cases, a small number of pre-flood images can be available/useful, due to low Sentinel-1 data availability, high precipitation rates or regular inundation patters (rice fields). In those cases, the assumption regarding the t-distribution of the backscattering coefficient will not be valid anymore. Exploiting more acquisitions with higher temporal separation from the flood date would increase the risk to introduce errors in *t*-score statistic due to land surface changes (removal of crop canopies). Future improvements of the developed toolbox would be related with the forming of a higher-quality baseline stack with support from multi-orbit Sentinel-1 data, ancillary data and data from other sensors. Exploiting multiple types of data would also lead to a contextualization of the flood extracted areas which is critical. Nevertheless, it is worth exploring since discrimination of the flooded areas can significantly help damage assessment and disaster response actions.

## 5. Conclusions

We described the methodological pipeline of FLOMPY toolbox and presented its practical use by analyzing Sentinel-1 time series for mapping floodwater. We demonstrated its robust performance using the same workflow with a fixed parametrization over several investigated cases around the world achieving high accuracy (OA~0.97, Kappa score~0.77). FLOMPY toolbox requires as inputs only the location and the time of the flood event along with the credentials for accessing Sentinel-1 and ERA-5 data. Using FLOMPY, even less-experienced users can exploit the Sentinel-1 time series in a standard and consistent way. The FLOMPY toolbox is open-source, modular and easy to integrate in workflows, therefore optimal for collaboration and potential future improvements. Future work is planned to exploit interferometric coherence information, other data sources and models in order to improve FLOMPY's performance and provide uncertainty information together with the flood product to better support hydrological modelling and disaster support decision systems. We consider this tool useful and relevant in the context of emerging initiatives such as EMS [46] and ARIA [47].

**Supplementary Materials:** The following are available online at https://www.mdpi.com/article/10.3390/w13212943/s1, Figure S1: Validation figure for EMSR504. (a) T-scores of the Sentinel-1 intensity dataset, (b) Flompy and EMS results overlaid over *t*-score image, Figure S2: Validation figure for EMSR456. (a) T-scores of the Sentinel-1 intensity dataset, (b) Flompy and EMS results overlaid over *t*-score image, Figure S3: Validation figure for EMSR497. (a) T-scores of the Sentinel-1 intensity dataset, (b) Flompy and EMS results overlaid over *t*-score image, Figure S4: Validation figure for EMSR520. (a) T-scores of the Sentinel-1 intensity dataset, (b) Flompy and EMS results overlaid over *t*-score image.

**Author Contributions:** Conceptualization all authors; methodology, K.K.; software, K.K.; validation, K.K.; formal analysis, K.K.; investigation, K.K.; data curation, K.K.; writing—original draft preparation, K.K.; writing—review and editing, all authors; visualization, K.K.; supervision, V.K.; project administration, V.K.; and funding acquisition, V.K. All authors have read and agreed to the published version of the manuscript.

**Funding:** This study was funded by the European Union's Horizon 2020 research and innovation programme under grant agreement No. 821054.

**Institutional Review Board Statement:** Not applicable.

**Informed Consent Statement:** Not applicable.

**Data Availability Statement:** The data presented in this study are available on request from the corresponding author.

**Acknowledgments:** We would like to thank the European Space Agency (ESA) for supplying the Sentinel-1 images. The authors would also like to thank the anonymous reviewers for their contribution to the improvement of the overall quality of the manuscript.

**Conflicts of Interest:** The authors declare no conflict of interest.

## Appendix A

FLOMPY toolbox exploits the dual polarimetric capabilities of Sentinel-1 via the product of VV and VH backscatter coefficients. The main reasons for this were (a) the decrease of computation cost because we work on one band instead of two and (b) the easier seperability of the product of VV and VH backscatter coefficients in respect to VV and VH backscatter coefficients seperately.

In the context of this paper, seperability is defined as the dissimilarity of the probability distribution function of the floodwater and non-floodwater populations. In Figure A1, the dissimilarity was calculated as the score of the two-sample non-parametric Kolmogorov-Smirnov test (KS). The product of VV and VH backscatter coefficients yields the highest KS score which is considered a good indication. This is showed in Figure 3 for the Pinios flood case study which presents mainly flooded low vegetated and bare soil plots. Furthermore,

FLOMPY's classification accuracy metrics for the same case study revealed the superior performance of the product backscatter coefficient.

- Single VV polarization: OA (0.967), UA (0.814), PA (0.704) Kappa score (0.738);
- Single VH polarization: OA (0.930), UA (0.911), PA (0.040) Kappa score (0.072);
- Product of VV and VH polarizations: OA (0.968), UA (0.783), PA (0.762) Kappa score (0.755).

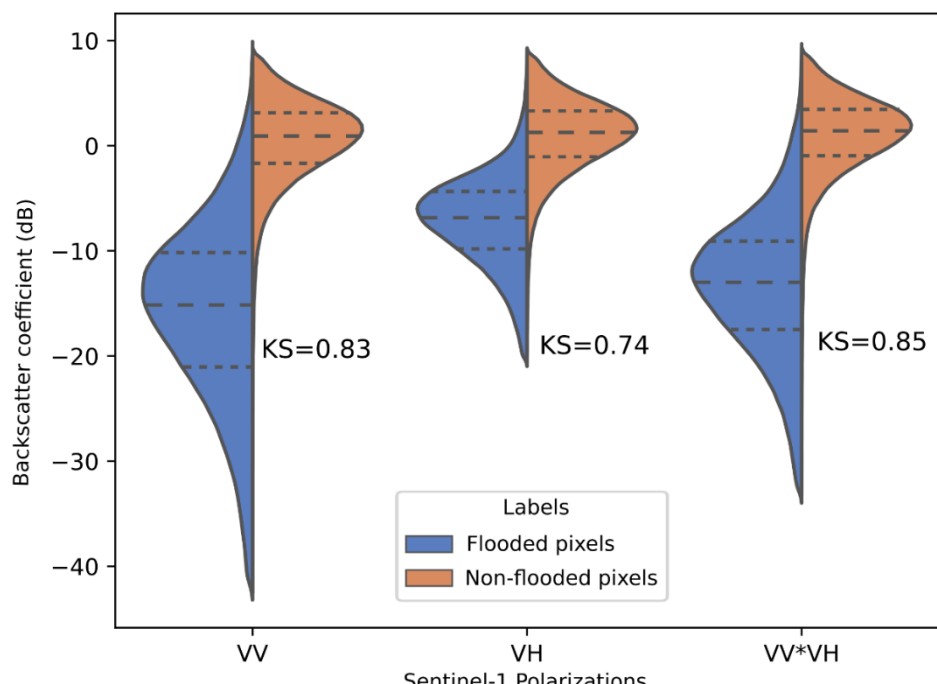

**Figure A1.** Distribution of backscatter coefficient for each polarization for flooded and non-flooded pixels for Pinios flood case study.

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
