# Peer review of "FLOMPY: An Open-Source Toolbox for Floodwater Mapping Using Sentinel-1 Intensity Time Series"

_water, doi:10.3390/w13212943_

Round 1

Reviewer 1 Report

The paper is exciting and is suitable for publication after minor revision.

  1. EMS product is regarded as the ground truth data in this paper. So what’s the difference between the methods adopted to produce your results and that of EMS?
  2. To enrich the applicability. It’s recommended that the authors find the surveyed flood extent data set and use it as the ground truth data to compare with the result of Flompy.
  3. Line 18, “overall accuracies were higher than 0.95”, what does this accuracy refer to?
  4. Would you please list the definition of those accuracies in lines254-257?
  5. Figure 7f) are recommended to thick the flood extent lines for Flompy and EMS. Or just left the two lines for comparison.
  6. Figure 6 shows optical Google earth imagery of other cases. In my opinion, it’s better to represent the results of Flompy and EMS.

Author Response

Please find my responses in the attached word file.

Reviewer 2 Report

The authors introduce a Python toolbox for floodwater mapping.  The article is interesting, and the paper is well-written.  I have a few comments that should be addressed to improve the presentation of this paper before publication.  The authors are to be commended for the release of open source software so this technology can be shared with other researchers.

line 8:  What is meant by “fully automatic”?  Please briefly describe using the results of the paper.

line 9: Indicate what is meant by “dense” in this context.

line 37: “From the other side” is unclear. Please re-write this sentence.

lines 55-56:  Consider re-writing these sentences to eliminate the word “lot.”  You can state that there are “many” drawbacks or “many” errors.

line 59: “Due to the continuous increase of the data volume and availability…” Is this data availability on the Internet?

line 87: What is the peak time of the flood event?  Is this the time coincident with the time of maximum discharge?

line 114: Why is “flood” listed in quotations here?  Please describe what is meant by the “flood” image.

line 200: Please insert a few sentences into the paper describing the Pinios flood event.  Therefore, lines 214-215 should be moved to the beginning of this section (Section 3.1).

lines 219-220: What is the context of this weblink?  Can a citation be provided instead of the weblink, or can the weblink be integrated into a citation?

lines 311-313: What is meant by “The high complexity of the scattering mechanisms”?  What makes the scattering mechanisms complex?  Please indicate here for context.

lines 334-335: “…less-experienced users can exploit the Sentinel-1 time series in a standard and consistent way” Why is this the case?  Please elaborate here in a few sentences.  Does the toolbox have a GUI or is the user guided in some way when using the toolbox?

Author Response

(The authors gave the same response as above.)

Reviewer 3 Report

I have attached my comments in the attached pdf file.

Author Response

(The authors gave the same response as above.)

Round 2

Reviewer 3 Report

I would like to congratulate the authors for addressing the comments.